# Regional and Seasonal Variability of Mineral Patterns in Some Organs of Slaughtered One-Humped Camels [*Camelus dromedarius*] from Saudi Arabia

**DOI:** 10.3390/ani12233343

**Published:** 2022-11-29

**Authors:** Mutassim M. Abdelrahman, Ibrahim A. Alhidary, Mohsen M. Alobre, Abdulkareem M. Matar, Abdulrahman S. Alharthi, Bernard Faye, Riyadh S. Aljumaah

**Affiliations:** 1Department of Animal Production, College of Food and Agriculture Sciences, King Saud University, Riyadh 11362, Saudi Arabia; 2CIRAD-ES, UMR SELMET, Campus International de Baillarguet, 34398 Montpellier, France; 3Department of Biotechnology, Faculty of Biology and Biotechnology, Kazakh National University Al-Farabi, Almaty 050013, Kazakhstan

**Keywords:** camels, elements, heavy metals, tissues, season, regions

## Abstract

**Simple Summary:**

Camel products are receiving great interest worldwide because of their high functional properties and nutritive values, but concern regarding their high content of unhealthy ingredients has also been raised. The trace, macrominerals, and heavy metal levels were investigated in the tissue and whole blood of slaughtered camels during the summer and winter seasons in Saudi Arabia. For whole blood, there was no significant effect of season on all mineral concentrations, but it was affected by regions in Saudi Arabia. A significantly higher Ca and P concentration was reported during the winter in liver tissues compared with the summer, with a significant variation in Ca, Mg, P, and Se between different locations. Furthermore, summer season affected the Cd and P concentrations in rumen tissues with no effect on rumen fluid.

**Abstract:**

Camel products are receiving great interest worldwide because of their high functional properties and nutritive values. Therefore, this study was focused on the variation of copper [Cu], zinc [Zn], manganese [Mn], selenium [Se], iron [Fe], iodine [I], and some heavy metals, cobalt [Co], lead [Pb], and cadmium [Cd], in the blood and tissues of slaughtered camels from five regions in Saudi Arabia [SA] during the summer and winter seasons, because environmental factors vary from region to region. Whole blood, meat, liver, rumen fluid, and rumen tissues were collected from the slaughterhouse in each region during the two seasons. Moreover, samples were prepared and analyzed for trace mineral and heavy metal concentrations using ICP-MS. The data were statistically analyzed as part of a complete randomized design and correlation analysis for season and location using SAS. The findings revealed a pattern in the minerals, with Ca being the only mineral that was unrelated to other minerals in the liver. For lead and cadmium, our mean value in liver [0.40 µg/g] was below the limit of the EU standard for cadmium [0.50 µg/g], while in meat and liver, lead contents [1.62 µg/g and 2.57 µg/g, respectively] were above the limit of the EU standard [0.10 and 0.20 µg/g, respectively]. For meat, the significantly highest positive correlations were observed between P and Mg [*R*^2^ = 0.928], Fe and Mn [*R*^2^ = 0.860], and Co and Mn [*R*^2^ = 0.821]. For rumen tissues, P and Mg were highly correlated [*R*^2^ = 0.958] as well as Zn and Mg [*R*^2^ = 0.857], Zn and P [*R*^2^ = 0.836], and Fe and Ca [*R*^2^ = 0.802]. As a result, a region and season reflect variations in mineral concentrations in SA during the summer and winter seasons. Further intensive research is needed to investigate the minerals’ biological mechanisms in camels under different environmental conditions.

## 1. Introduction

The demand for animal products in Saudi Arabia [SA] as well as in other southern countries is increasing rapidly, but the supply is still below the required level, in part because of low animal productivity. Camel [*Camelus dromedarius*] is considered one of the main sources of healthy meat, milk, and liver for human consumption within SA [1]. The number of camels in SA fluctuates from year to year [2] because of variations in conditions caused by drought, erratic rainfall, and global warming. Camels suffer from growth retardation in newborns, low feed efficiency, anemia, poor fertility, poor reproduction, and many other metabolic disorders, especially with the intensification of camel farming. In such context, trace mineral deficiencies, toxicities, and heavy metals can occur more often and negatively influence camels’ production and reproductive efficiency, as well as many aspects of growth, metabolism, and health [3].

Minerals [macrominerals, electrolytes, and trace elements] are crucial for animal health and productivity, including camels, especially when they become a limiting factor of the diet [4]. They play a pivotal role in many physiological activities, and their deficiency causes a variety of pathological problems and metabolic defects [5]. The levels of nutrition and trace mineral intakes are known to affect the reproductive ability of male and female camels [6,7]. The trace minerals such as selenium [Se], copper [Cu], zinc [Zn], manganese [Mn], cobalt [Co], iron [Fe], iodine [I], and molybdenum [Mo] are involved in the normal growth and productivity of livestock [8]. Infertility, non-infectious abortion, anemia, and metabolic diseases are some of the main clinical signs of deficiencies and abnormalities [9,10]. A few scientific studies have shown some evidence of the sensitivity of camels to trace mineral disorders because of either deficiency or toxicity in the same way as other ruminants [4,11]. Faye and Bengoumi [11] and Liu et al. [12] have reported several incidences of clinical mineral deficiencies in camels that were underestimated because signs of subclinical deficiencies may remain undetected for long periods. Regarding toxicity, evidence is even rarer. For example, some cases of selenosis [13] as well as fluorosis [14] were described.

Heavy metal concentrations in camels and other ruminant animals are an important issue since they negatively affect human health, according to the WHO [15]. Toxic metals such as lead [Pb] and cadmium [Cd] can be found in contaminated food from animals and plants that are produced in environmentally polluted areas [16]. Lead and cadmium cause many biochemical dysfunctions and many fatal diseases in humans and animals [17,18]. The monitoring of heavy metals in camels’ tissues can provide critical information regarding the pollution [19,20]. Furthermore, heat stress during summer can lead to negative mineral balance as a result of decreasing feed intake, digestion, and consequently increased mineral excretion [21] in Saudi Arabia. Therefore, the present study was designed to evaluate the mineral status of camel’s tissues in slaughtered animals [meat, liver, whole blood, rumen fluid, and rumen tissues] from five different regions in Saudi Arabia during two seasons.

## 2. Materials and Methods

### 2.1. Ethical Approval

The research was carried out in accordance with the Kingdom of Saudi Arabia’s ethical standards for animal use (Ethics Committee, King Saud University, Ethical (approval number: KSU-SE-22-21). The camels used in this study were slaughtered in a slaughterhouse for meat production.

### 2.2. Regions and Temperatures

This study was conducted in five selected regions in Saudi Arabia [Figure 1]: A. Central Region [Al Riyadh: 50 samples]; B. Western Region [Mekka: 62 samples]; D. Eastern Region [Dammam: 32 samples]; C. Southern Region [Najran: 13 samples]; and Northern region [Al-Jouf: 19 samples]. The mean temperature was 41.6 + 3.4 °C, with a low variability between regions [Figure 2]. Soil and water contents in trace minerals and heavy metals, for each region [Table 1 and Table 2], also varied, with lower values for most of them reported in the southern and northern parts of Saudi Arabia. This may have an impact on the regional mineral status of the animals grazing in those different areas. Soil, water, and feed are the main sources of dietary minerals for camels raised in semiarid and arid environments. However, in summer, because of the limited range of plants, the dietary regimes were based on alfalfa hay [*Medicago sativa*] and Rhodes grass [*Cenchrus ciliaris*] in all regions. Therefore, in summer, the dietary mineral supply in the five regions under investigation was probably less variable than in winter.

### 2.3. Sampling Procedure

Only male camels > one year old (up to 2 years) were selected in the regional slaughterhouses to be as homogeneous as possible for sample collection. In total, 107 samples were obtained in the summer and 69 in the winter [Table 3]. Sampling was performed in two seasons [summer/winter] in the five regions [Table 3]. It included whole blood [16 samples], liver [73 samples], and meat [53 samples]. Rumen fluid [20 samples] and tissues [14 samples] were collected in only two regions [the central and eastern regions]. Whole blood and tissue samples were stored at −20 °C until further analysis.

### 2.4. Laboratory Work

All samples were prepared and digested thoroughly using acid wet ashing [23] before being analyzed for trace mineral concentration using ICP-MS. Whole blood, rumen fluid, tissues, liver, and meat were prepared as follows: 0.500 ± 0.001 g of tissue samples—liver, meat, and rumen tissues were weighed in an acid-washed Teflon vessel. Then, 3 mL HNO3 [65% Riedel-de Haen, Munich, Germany], 1 mL HCl [36% Avonchem, Macclesfield, UK], 1 mL H2O2 [30% *w*/*v* Avonchem, Macclesfield, UK], and 1 mL deionized H2O [Milli-Q quality] were added to the sample before loading on the digestion units. The samples were digested according to a pre-set temperature program as recommended by the AOAC [23]. The digested samples were diluted in a 25 mL volumetric flask using 0.1 normality HCl and mixed very well. Subsamples [5 mL] were taken in sterilized tubes for mineral analysis.

### 2.5. Minerals Analysis

Determinations of the trace minerals were carried out using ICP-MS equipped with a Meinhard Nebulizer type A2. Argon, [purity higher than 99.999% and supplied by the AH Group, Dammam, Saudi Arabia], was used to sustain plasma and as a carrier gas. The operating conditions employed for the ICP-OES determination were 1300 W RF power, 15 L min^−1^ plasma flow, 0.2 L min^−1^ auxiliary flow, 0.8 L min^−1^ nebulizer flow, and 1.5 mL min^−1^ sample uptake rate. Metals were determined using axial and radial views, while the analyte signal was measured using two-point background correction and three replicates, with the peak area processing mode. The emission intensities were obtained for the most sensitive lines, free of spectral interference. The calibration standards were prepared by diluting the stock multi-elemental standard solution [1000 mg L^−1^] in 0.5% [*v*/*v*] nitric acid. The calibration curves for all elements were in the range of 1.0 ng mL^−1^ to 1.0 µg mL^−1^ [1–1000 ppb].

### 2.6. Statistical Analysis:

The objectives of the statistical analyses were:To identify the effect of variation factors [season and region] on the mineral status of the organs and blood;To characterize the mineral patterns [profiles] for a given organ and blood;To assess the relationships between minerals according to their concentration in all organs and blood;To assess the relationships between the mineral patterns and the variation factors;To assess the correlations between organs and blood for each region.

For achieving those objectives, different statistical procedures were used [24]: for objective [1], a variance analysis [ANOVA] was applied by using season and region as independent variables and normalized mineral values as dependent variables; for objective [2], we used a multivariate analysis [Principal Components Analysis-PCA] followed by an Ascending Hierarchical Classification [AHC], with each class identified corresponding to a specific mineral pattern; for objectives [3 and 5], the data were submitted to correlation analysis using the Pearson correlation coefficient; and for objective [4], a table of contingency crossing the mineral patterns [classes issued from AHC] and the independent variables [region, season, and later organs] was implemented and a chi-square test was applied to characterize their relationships. Level of significance was *p* < 0.05. The statistical software used was XLstat [25].

## 3. Results

### 3.1. Main and Trace Mineral Concentrations in Different Camels’ Tissues

The inter-organ differences in mineral concentrations were significant for all elements [*p* < 0.0001], except for lead [*p* < 0.003; Table 4]. Regarding main minerals, the most abundant in all organs is phosphorus [Figure 3a], where it represents between 75.1% [liver] and 90.9% [blood], except in rumen fluid, where magnesium is slightly more abundant than phosphorus [47.9% vs. 45.2%]. Among trace elements [Figure 3b], copper is dominant in rumen tissue [61.0%], liver [41.0%], and meat [35.3%], while iron is in higher proportion in blood [49.3%] and rumen fluid [27.7%] just before copper [26.6%]. The third abundant trace element is zinc, notably in meat [26.4%]. In blood, zinc is the second element [19.0%] before copper [15.4%]. Rumen fluid contains the highest proportion of manganese compared to other organs [18.0%] as well as iodine [4.2%]. The higher proportion of cobalt is in whole blood [1.2%] as well as selenium [9.7%]. Although the concentration of lead is higher in the liver, the organ with the highest proportion is the rumen fluid [1.7% of the total trace elements].

### 3.2. Seasonal and Regional Effects

Due to the high variability in organ mineral patterns, seasonal and regional effects were assessed for each organ separately.

#### 3.2.1. Liver Samples

There was no significant seasonal variation for all elements except calcium and phosphorus, which were higher in the winter [Table 5]. Regarding regional variability, calcium and manganese were in higher concentration in camel liver at Dammam, while phosphorus and selenium were higher at Al-Jouf. Magnesium was in higher concentration at Najran. No significant values were observed for the other elements [Table 5].

#### 3.2.2. Meat Samples

If the seasonal variability in liver samples involved the main minerals [Ca, P], in meat samples, only cobalt and iodine slightly differed between summer and winter, with a significantly higher mean value of cobalt in winter and of iodine in summer [Table 6]. In contrast, the regional origin of the meat samples had a significant effect on most of the minerals, except calcium, copper, and lead. The Riyadh region was characterized by low levels of cobalt and high levels of iodine and magnesium. The Mekka region is the lowest in iron, iodine, manganese, and zinc. Meat samples from Dammam were poor in magnesium. Cobalt, iron, manganese, and zinc were in higher concentrations in the meat samples from Najran, while selenium was very low. Meat samples from Al-Jouf contained higher concentrations of selenium. Moreover, a significant variation (*p* < 0.0219) was reported for Cd concentration in camels’ meat between summer and winter (0.159 vs. 0.271 µg/g ww). Camels raised in Najaran reported a significantly higher level of Cd in their meat compared with other regions [Table 6].

#### 3.2.3. Rumen Tissue

Except for cadmium, there were no significant seasonal differences between winter and summer regarding mineral concentrations in rumen tissue [Table 7]. Between the two regions involved in rumen tissue sampling, highly significant differences were observed for magnesium, phosphorus, and zinc [Table 7], with the highest values for these three minerals being reported in the eastern region [Dammam].

#### 3.2.4. Rumen Fluid

Contrary to rumen tissues, there were no significant seasonal and regional differences in the mineral concentrations in rumen fluid, whatever the minerals [Table 8].

#### 3.2.5. Whole Blood

For whole blood, there was no significant effect of season on mineral concentrations in blood samples. On the contrary, calcium, magnesium, and phosphorus [the main minerals], as well as selenium and zinc, were higher in the Dammam region than in the Riyadh region, while cobalt, copper, iodine, and manganese were lower [Table 9].

### 3.3. Mineral Patterns and Correlations in the Different Organs and Fluids

The PCA followed by AHC presented two types of main results: a correlation circle allowing assessment of the relationships between minerals within a specific organ and a typology of homogenous mineral patterns that could be explained by the independent variables, season, and region.

#### 3.3.1. Mineral Patterns in Liver Samples

In the liver, only calcium was not correlated to any other minerals. In reverse, regarding the main minerals, magnesium was positively correlated to copper and cadmium and negatively correlated to selenium and phosphorus. In addition to selenium, phosphorus was also positively correlated with iodine [the highest correlation coefficient: *R*^2^ = 0.673; *p* < 0.0001], manganese, and zinc [Figure 4a]. Regarding trace elements, only one negative correlation was observed [Co/Zn, *R*^2^ = −0.245; *p* < 0.05]. Positive correlations were revealed in some cases: Co/Fe, Co/Mn, Fe/Mn, I/Se, I/Pb, and Mn/Se [Figure 4a].

After classification, three patterns were identified and represented on the main factorial plan [F1, F2]. The three patterns expressed 67.2% of the total variance [Figure 4b] and can be described as follows [Table 10]:L1 [on the left-downside of the factorial plan in Figure 4b] was characterized by significantly lower concentrations in Co, Fe, I, Mn, P, Se, and Zn and medium values for Mg. Those patterns were not explained by season and region, although some livers collected at Al-Jouf and Dammam were particularly low in most of the minerals;L2 [on the right side] was low in Co and Mg, but at the reverse end it contained the highest concentrations of I, P, Se, and Zn. There was no seasonal or regional effect, although liver samples from Al-Jouf region were more abundant in this group;L3 [on the left-upside] contained very high values in Co, Fe, and Mg but low values in Se and Zn without seasonal or regional effect.

#### 3.3.2. Mineral Patterns in Meat Samples

The highest positive correlations were observed between P and Mg [*R*^2^ = 0.928; *p* < 0.0001], Fe and Mn [*R*^2^ = 0.860; *p* < 0.0001], and Co and Mn [*R*^2^ = 0.821; *p*< 0.0001]. Other positive correlations occurred, mainly for calcium [Ca/Co, Ca/Fe, Ca/Mg, Ca/Mn, Ca/Zn] and zinc [Zn/Co, Zn/Fe, Zn/Mg, Zn/Mn]. The only significant negative correlation was between Se and Co [*R*^2^ = −0.337; *p* < 0.05]. However, contrary to what was shown for liver samples, as all minerals were projected to the same side of the correlation circle (to the right side) [Figure 5a], the tendency was an increasing mineral concentration in the same way with higher mineral content in meat from Al-Jouf or Najran [Figure 5b]. Three other patterns were also identified after classification, explaining 61.7% of the total variance. These three groups of meat samples could be described as follows:M1 [left-downside of the factorial plan in Figure 5b] was used to group meat samples with low mineral concentrations for all elements [Table 11]. There was no seasonal effect, but the samples from Dammam were significantly more abundant in this pattern [*p* < 0.001];M2 [upside] correlated with meat samples rich in selenium, phosphorus, and magnesium. In addition, iodine was also significantly higher in concentration, and zinc was medium. The samples from the Al-Jouf region were more abundant in this group [*p* < 0.01], while Najran and Dammam were significantly less represented [*p* < 0.05];M3 [right-downside] was gathering the richest samples in trace elements [Co, Fe, Mn, and Zn] and calcium. In reverse, selenium content was the lowest. The samples from Najran were significantly more abundant in this group [*p* < 0.001].

#### 3.3.3. Mineral Patterns of Rumen Tissues

Phosphorus and magnesium were highly correlated in rumen tissue [*R*^2^ = 0.958; *p* < 0.0001] as well as zinc and magnesium [*R*^2^ = 0.857; *p* < 0.0001], zinc and phosphorus [*R*^2^ = 0.836; *p* < 0.0001], and iron and calcium [*R*^2^ = 0.802; *p* < 0.001]. Other positive and significant correlations were observed for Cd/Mg, Cd/Mn, Cd/Fe, Cd/I, Co/Mn, Fe/Mn, and Cu/Se. Usually, no correlation was found with Pb in the liver or meat, but in our rumen tissue samples, there were slight but significant correlations with Cu and Mg [*p* < 0.05]. There was no negative correlation, and the correlation circle [Figure 6a] shows that all elements are never increased in opposition and are linked mainly to the first factor. No seasonal effect was observed. Only two patterns explaining 71.6% of the variance were identified after classification [Figure 6b and Table 12]:RT1 [right-downside] was mainly observed in the Dammam region [*p* < 0.0001] and gathered samples with significantly lower Mg, P, and Zn. Other elements were not significantly different between the two patterns;RT2 [left-upside] samples of rumen tissue rich in those same elements were represented in reverse. Thus, Mg concentration was 3.5 times higher in RT2 compared to RT1, and P concentration was 2.9 times higher. Those samples are mainly from Riyadh [*p* < 0.0001].

#### 3.3.4. Mineral Patterns of Rumen Fluid

Three correlations were highly significant [*p* < 0.001], i.e., Ca/Mn, Fe/Mn, and P/Mg [like for rumen tissue]. There was no significant negative relationship between the elements as in the previous substrate [Figure 7a]. Other significant correlations occurred for Ca/Cd, Ca/Fe, Ca/Mg, Cd/Se, Cd/Zn, Fe/Cd, Fe/Se, Fe/Zn, I/Cd, Mg/Co, Mg/Fe, Mg/Mn, Mg/P, Mg/Pb, Mg/Se, P/Mn, and Se/I. The two patterns identified after classification, named RF1 and RF2 [Figure 7b; Table 13], could be described as follows:RF1 [left side] had rumen fluids significantly lower for the main minerals [calcium, magnesium, and phosphorus] and manganese;RF2 [right side] was characterized in reverse to high level for the same elements.

The projection of the supplementary variables [region and season] in the factorial plan close to the center of gravity of the factorial plan [Figure 7b] was indicating a lack of regional or seasonal effect.

#### 3.3.5. Mineral Patterns in Whole Blood Samples

Contrary to the other substrates, there were several significant negative correlations [Figure 8a]. Globally, phosphorus, magnesium, and selenium had opposite concentrations to trace element concentrations such as copper, cobalt, manganese, and iodine. The highest positive correlations were Mn/Co [*R*^2^ = 0.967; *p* < 0.0001] and Mg/P [*R*^2^ = 0.835; *p* < 0.0001]. Other positive correlations were observed between Ca/Mg, Cd/Fe, Co/I, or Mg/Ca.

Three patterns explaining 86.5% of the total variance were identified following the classification of the whole blood samples [Figure 8b and Table 14]:WB1 [right-downside] included only two samples from the Riyadh region and was characterized by high values in trace elements [Co, I, Mn, and Zn] and low values in main minerals [Mg, P];WB2 [center] included the majority of samples with medium values for all the significant minerals [Co, I, Mn, Mg, and P], except Zn in the lowest concentration;WB3 [left side] included only three samples, with a pattern reversed from WB1, i.e., rich in main minerals [P, Mg] and poor in trace elements [Co, I, and Mn] except Zn.

The projection of supplementary variables supported the hypothesis of a regional effect, with the Riyadh region being close to WB1, while Dammam was closer to WB3. In reverse, there was no seasonal variability.

#### 3.3.6. Correlations between Regional Matrices

There was no correlation for the different minerals between the matrices [soil, water, liver, and meat] except on two occasions: [i] liver and meat Se contents were significantly correlated [*R*^2^ = 0.948; *p* < 0.01] and [ii] soil and liver Co also [*R*^2^ = 0.917; *p*< 0.05]. However, no relationship between the values of trace elements in soil and water was observed. As only two regions were investigated for other organs, and as main elements were not determined in soil or water, it was not possible to test such relationships.

#### 3.3.7. Correlations between Organs

The correlation matrix was built by taking into account the mean of each organ’s values, as the number of samples was not the same. The mean mineral contents of one specific organ was correlated to all other organs (Table 15).

On the other side, each mean mineral value, whatever the organ, was also correlated to the other minerals, except for calcium, whose concentration is not correlated to other minerals, and zinc, which is not correlated to Mn, I, and Cd.

## 4. Discussion

The level of minerals in different tissues could be regarded as an indicator of mineral status in animals even if, in practice, mainly serum or plasma are investigated, including camels [3]. The mineral patterns could also give additional information about relationships with the levels of elements provided by the diet in specific environments. The first observation is the lack of significant seasonal effects on mineral patterns, whatever the substrate. Only a few minerals appeared significantly different [generally with probability *p* < 0.05] between winter and summer: calcium and phosphorus in the liver, cobalt and iodine in meat, and cadmium in rumen tissue. However, even if the differences were statistically significant, the biological significance was of low interest because the values were within normal concentrations, i.e., not in excess or deficient [3]. At reversal, regional variability appeared important [except for rumen fluid] for individual minerals [for example, liver calcium concentration, which was five times higher in Dammam than in Najran], but this regional variability was less marked when full mineral patterns were considered.

### 4.1. Mineral Pattern in Camel Liver

Relatively few references were available regarding the main minerals in camel liver. In previous data from Awad [26], the calcium concentration was 927 ppm (µg/g), i.e., in proportion ten times higher than our results. More convenient values were published by Dawood and Alkanhal [27], with an average of 47 µg/g. Calcium in the liver is not stored passively but is a versatile secondary messenger that regulates multiple hepatic functions, especially in cases of injury [28]. Thus, the regional variability could be linked more to the health status of the liver than to the diet of animals. Magnesium was also rarely determined in camel liver: 220 ± 60 µg/g [29] or between 702.6 and 807.8 µg/g according to different camel breeds in Saudi Arabia [30]. As for calcium, magnesium status in the liver is rather linked to liver integrity [31] than to passive storage. Liver is particularly rich in phosphorus, with concentrations exceeding 40 mg/g. In the reference of Awad and Berschneider [26], a similar amount to ours was reported [6382 ppm, i.e., 63.82 mg/g], and more recently, Mustafa et al. [32] reported a range from 30 to 80 mg/g according to the camel gender. In the three mineral patterns of camel liver, there was no difference in calcium, and a reverse relationship occurred between magnesium and phosphorus, with the highest concentration of P being associated with the lowest concentration of Mg. Many complex interactions between Ca, Mg, and P occurred in ruminants [33], and their levels in the liver are more reflective of the organ’s integrity status than the mineral composition of the diet.

Contrary to the main elements, some trace elements can be stored in organs, such as the liver. It is notably the case of copper for which the liver concentration is reflecting the quantity of copper provided by the diet [Roberts and Sarkar, 2008]. For example, in dromedaries, copper supplementation in the diet [9.5 g of copper sulphate, daily] multiplies hepatic copper concentration by three [34]. By using boluses containing copper, Ibrahim et al. [35] increased the copper content in camel liver from 19.7 to 48.7 µg/g. Although no significant differences occurred between regions in our study, some values were below 75 µg/g (Al-Jouf and Najran), regarded as critical levels in the liver [36]. In Bactrian camels, low copper concentrations in the liver [14 ± 3 µg/g] were recorded in animals affected by the “emaciation ailment” [37]. In camel livers collected in Sudanese slaughterhouses, a mean value of 103 ± 12.3 µg/g was reported [38]. However, in Iran, the content was much lower [3.28 ± 0.79 µg/g] [39]. In Iraq, a seasonal variation was reported between 4.66 and 26.75 µg/g [40]. In their study comparing camel and cow receiving similar diet and mineral supplement, Bengoumi and Faye [41] found lower content in camel hepatic copper (9 µg/g) than in the cow (35 µg/g) before supplementation, with a similar gap after mineral supplementation (32 vs. 109 µg/g). Finally, the critical limit of 75 µg/g proposed above for cows seems not applicable for camels. Moreover, in comparison to the literature data [range of 13.6–163 µg/g] in Faye and Bengoumi [3], our results appeared widely above any potential critical limit.

Contrary to copper, zinc is not easily stored in the liver. In the literature, liver zinc concentrations varied from 11 ± 1.7 to 309 ± 78 ppm [3], although lower values were reported recently by El-Ghareeb et al. [42] (3.96 ±1.02 ppm) and Morshdy et al. [43] (6.16 ± 0.67 ppm). The link between those values and deficiency or toxic status was unclear, especially because the type and quality of analytical procedures to determine Zn could vary between studies. Moreover, it seems that per-os Zn supplementation with Zn sulphate did not significantly increase Zn content in the liver [44] nor did protéo-energetic supplementation with trace elements [4]. By using mineral boluses, however, Alhidary et al. [35] obtained a slight increase in liver Zn storage from 31.3 ppm in the control group to 42.8 ppm with one bolus and to 52.1 ppm with two boluses. There was no evidence of regional variability potentially linked to feeding variability in our investigation. Even between the three liver patterns, although there was a significant difference in Zn concentration, the variability appeared narrow (from 49.72 to 66.36 ppm), which is biologically insignificant. In the literature, higher variability was reported between types of camels, for example, between racing and pack camels 163 ± 64 and 224 ± 86 ppm, respectively [45]. The same author did not find significant differences at the slaughterhouse between Zn concentrations in young or adult camels of 181 ± 58 and 182 ± 30 ppm, respectively. However, a breed effect was reported in Saudi Arabia [30], with a double concentration in the Maghateer breed (89.9) compared to the Majaheem breed (43.7 ppm). On 60 camel livers from Iran, no regional or seasonal differences were also observed, but a slight sex difference was reported, with the concentrations in males (mean = 121; range 73–155 ppm) being higher than females (mean = 103; range 71–155 ppm) [39]. In contrast, an important seasonal variation was revealed by Al-Perkhdri [40] in Iraq, with Zn concentrations being quite higher in the summer (103.8 ± 6.1) than in the spring (61.9 ± 1.5 ppm).

In the literature, liver iron contents were usually between 300 and 600 ppm [3], except in some unexplained cases where they overpassed 2500 ppm [41] or, at the reverse, were below 20 ppm [42]. In this context, our results appear low (around 100 ppm). There was no significant seasonal or regional variability in our study, but an important difference occurred between patterns. The lack of a seasonal or regional effect was also observed by Asli et al. [39]. According to data from previous research by Tartour [46], a negative correlation between iron and copper concentrations in camel liver was reported and attributed to the competition for storage sites in the hepatocytes. However, it was not confirmed in our case, although a similar hierarchy of values was observed in the three types of liver: the lowest Cu and Fe concentrations were in pattern L1, reversing to pattern L3 (Table 10). Usually, desert plants are not iron-deficient. However, in the case of chronic liver disease, which could occur in camels, iron regulation may be disturbed, and high iron-levels could be observed in cases of viral infection or metabolic disorders. Such liver disorders could alter the synthetic functions of the liver, notably by decreasing the production of hepcidin, a key protein in iron metabolism, resulting in iron deposits in the liver [47]. Even if camel hepcidin was characterized [48], the link between this protein and liver iron status was never explored to our knowledge.

Hepatic manganese concentrations in the liver range between 2 and 10 ppm [3,49] and are comparable to other species [49,50,51]. However, lower values were recently reported with a range of 0.50–1.23 ppm [40]. With a mean of about 24 ppm, our results appear higher. Even in the case of Mn supplementation by rumen bolus [52], the Mn concentration did not exceed 5 ppm. In contrast to Asli et al. [38], no seasonal effect was observed in our study, but a regional variation was present. Because the liver is a manganese storage organ, the high level in our study could be linked to a relatively high level in desert plants in the region, even though the amount of Mn in the soils did not appear to be particularly abundant. Mn intoxication could lead to liver failure [53]. The association in our study of low Fe and high Mn in our samples could point to suspicion of subclinical liver failure associated with parasitism [3].

Liver selenium levels appeared to be highly variable between regions, with a ratio of 8:1 between Al-Jouf and Najran areas. Selenium deficiency in camels was widely observed in the Arabian Peninsula, with soil and forages being regarded as deficient [54] in many places. Indeed, Se concentrations in soil samples from the northern region [Al-Jouf] were 6.5 times higher than in the southern region [Najran]. However, few references are available regarding hepatic selenium concentration [3]. In comparing the quantity of selenium in different organs, Seboussi et al. [55] found lower concentration in the liver than in the kidney in non-supplemented camels with Se, while it was the reverse in supplemented groups. However, on average, there was more Se in the kidney than in the liver, and the quantity in the last organ varied between 0.22 and 1.51 µg/g, i.e., quite lower values than the present report. In Bactrian camels, a range of 1.1–1.5 µg/g was also observed [56]. Thus, in view of the results, no deficient status can be detected in our samples, whatever the region. The differences between liver patterns were also marked, and a high selenium level (pattern L2) was linked to high phosphorus, zinc, and iodine but low cobalt and magnesium. The literature describes numerous complex interactions between minerals in the liver. For example, in pigs, it has been reported that dietary phosphorus levels have a positive effect on Se retention and liver Se storage [57]. A positive correlation between liver zinc and selenium was also described in humans [58].

If iodine and cobalt did not present seasonal or regional variation, they contributed to differentiating the liver patterns, especially cobalt, which, in connexion with iron, magnesium, and manganese, was particularly high in the pattern L3 (almost ten times more than in other patterns).

In the literature, liver cobalt varied considerably according to authors [all data converted into µg/g for a better comparison]: 0.057–0.063 [59], 0.68 ± 0.21 [7,37], 0.81–0.89 [56], 0.6–2.0 [37], 1.87 ±0.35 [60], 2.0 [37], and 2.46–3.93 according to supplementation status [34]. Thus, the pattern L3 appeared considerably high in cobalt. The values of cobalt in meat were reported to be comparable to those reported by Badiei et al. reference: 0.002–0.089 µg/g [61]. However, in Bactrian, Raiymbek et al. [62] reported higher values: 20–40 µg/g DM in different muscles, i.e., approximately 12–24 µg/g wet.

To our knowledge, there was no reference regarding iodine content in camel liver, and therefore, no critical limit reported in the literature for iodine to compare with our results. However, there was no sign of goiter in the camels raised in all regions, contrary to more continental areas [50]. In beef meat, quite lower values were reported: 0.038 ±0.01 µg/g [63]. In another study, the iodine content of various meats [except camel] ranged from 0.71 to 2.16 µg/g, with an average of 1.44 ± 0.28 µg/g [64].

### 4.2. Mineral Pattern in Meat

If the seasonal variability of camel meat mineral composition was limited to iodine and cobalt, the regional variability involved almost all the elements except Ca, Cu, and Pb [Table 6]. The same elements contributed to differentiating the meat patterns, in addition to calcium. However, remarkable high concentrations occurred only for cobalt-manganese [pattern M3].

With values reported in the literature between 125.6 ± 17.8 [65] and 270 [66], or with ranges of 92–466 [67] and 133–243 µg/g [52], our values were within the normal level of calcium in camel meat.

Phosphorus is the most abundant mineral in meat. As for calcium, the regional variability was limited, although statistically significant. In any case, our results are comparable to those reported by Kadim et al. [68] [2499–5840 µg/g], [66] [5490 µg/g], Raiymbek et al. [62] [2290 ± 670 µg/g] in Bactrian camels, [69] (176.0 ± 4.30 µg/g) and Ibrahim et al. [52] (3520–4120 µg/g). The highest phosphorus in pattern M2 was not necessarily linked to exceptional high or low levels of other elements, except selenium. Negative relationships between phosphorus and selenium were described in pigs, where dietary P levels could reduce the absolute and percentage of selenium retention [57].

Comparable values with those in the literature regarding meat magnesium were also reported. For example, 560 µg/g [66], a range from 247 to 573 µg/g [67], and from 356 to 444 µg/g [51], from 901.6 ± 50.3 µg/g [64], i.e., more than in other species. A variability in breed was observed in Saudi Arabia [30], which could partly explain the observed regional differences.

Regarding trace elements, less coherence is observed in published data. For example, for copper, with a mean of 67.05 µg/g, our results are quite higher than those of Abdelrahim et al. [69] (14.4 ± 3.3), El-Faer et al. [70] (0.7–0.9 µg/g), or Hammad et al., [71] [1.6 g/g], but lower than those of El-Ghareeb et al., [42]: 150 ± 20 µg/g. A very low level of copper in muscle was also reported by Morshdy et al. [43]. The same is true for zinc, where our results appeared quite higher 9.84 ± 0.36 µg/g for Chafik et al. [72] in Morocco; 1.16 ± 0.52 µg/g for El-Ghareeb et al. [42] in Egypt; 8.35 ± 1.33 µg/g for Morshdy et al. [43] again in Egypt; similar [46 ± 2.2 µg/g for Abdelrahman et al. [30] in Saudi Arabia, 34.9 ± 0.02 µg/g for Hammad et al. [71] in Sudan or quite lower [179.3 ± 48.7 µg/g for Liu [73] in Chinese Bactrian camel. In their recent publication, El-Boukhary et al. [74] stated that on average, copper and zinc in Mauritanian camel meat were 6.8–9.4 µg/g and 27.9–38.6 µg/g, respectively. No significant variation was reported for zinc between muscles [75]. The variability observed in our survey could not be attributed to a potential difference in muscle samples. The role of zinc in skeletal muscle performance and resistance to fatigue could explain the relative importance of this element in muscle.

Iron is also an important element for muscle performance. The iron concentration in muscle was comparable to that of zinc, with significant regional variability. Iron levels in meat have been reported in the literature to range from 28.9 to 61.0 µg/g [26,38,76], with differences between muscle types [70]. In some references, slightly lower values were reported: 10.9–21.0 µg/g [42], 19.4 µg/g [26], and 25.3 µg/g [59]. Higher differences could occur in the literature, notably due to the expression of the results [reported or not on dry matter] and to analytical procedures. Anyway, our meat samples fell within the normal range, whatever the region. In the meat pattern M3, iron was particularly high [80 µg/g], similarly to other main trace elements such as Cu, Zn, Co, and Mn. Most of the samples from this pattern came from the Najran region, a mountainous region that may necessitate more muscular effort to occupy a space defined by its reliefs.

Selenium was in lower concentration in meat from the Najran region, which is also the place where selenium in the soil appeared to be the lowest. It is also in pattern M3 that the Se concentration is quite low [3.68 µg/g] compared to other patterns; the reverse is true for all other trace elements. In the experiment of Seboussi et al., [55], muscle Se concentration varied between 0.35 to 0.42 µg/g with an effect of Se supplementation in the diet. In a recent study [50], the concentration of meat selenium varied from 0.23 ± 0.05 [in shoulder muscle] to 0.41 ± 0.41 µg/g [in thigh muscle]. Based on the literature references, our results appeared to be very high.

The regional variability of manganese was also remarkable, ranging from 2.94 to 13.07 µg/g. Those values are comparable to the range of 1.2–2.5 µg/g reported in China for Bactrian camels [56,73], but quite higher than results obtained by other authors in Dromedary camels [30,70], with values widely below 0.5 µg/g. The manganese concentration was high in the pattern M3, as were the other trace elements except selenium.

Cobalt is a rare trace element, with iodine presenting a seasonal variation. Moreover, the regional variation was also important, with values ranging from 0.32 µg/g in Riyadh to 2.07 µg/g in Najran. All the meat samples with high Co values were gathered in pattern M3, which had the richest profile regarding trace elements. Our mean result [1.15 µg/g] was in accordance with that of Ma [56]: 1–1.2 µg/g, or 0.6–2 µg/g [73], but lower values were reported by Badiei et al. [59]: 0.043–0.052 µg/g. The concentration of cobalt in muscle varied from 1.7 to 2.8 μg/g [75].

Iodine was not investigated in camel meat. In our results, the iodine content in camel meat was similar all over the country. Despite a slight statistical difference between meat patterns, there was no apparent biological significance.

### 4.3. Mineral Patterns in Rumen Tissue

The rumen wall is an important route for the absorption of nutrients present in the rumen fluid [77], including minerals. The mineral content of rumen tissue could reflect the mineral patterns of rumen fluid, i.e., of the diet [78]. Notably, the iron content of cows’ diet has been shown to influence the dark-brown color of their rumen epithelium [79]. In our samples, manganese, selenium, and cadmium in rumen tissue were highly correlated to their concentrations in rumen fluid. However, contrary to the liver or muscle, the rumen wall is more an organ of nutrients’ flow than storage. To our knowledge, there is no data regarding the mineral composition of rumen tissue. An important regional difference was observed for magnesium and phosphorus, mainly with values that were three times higher in Dammam compared to the Riyadh region. This regional variability in these two minerals explained the two mineral patterns, which were characterized by a contrast between rumen tissue rich in Mg and P [pattern RT1] and those with low values. Zinc was also a discriminating parameter of the mineral patterns [and of the two regions], but to a lesser extent. Such a high difference between regions could be linked to mineral supplementation practices, but information regarding the type of diet is lacking. Furthermore, no such difference was found between these two regions in meat and only for magnesium in liver.

### 4.4. Mineral Patterns in Rumen Fluid

The mineral concentration in the rumen fluid gives a good indication of mineral digestion and availability during ruminal microbial fermentations. Yet, there was no regional or seasonal effect on the mineral composition of rumen fluids in our samples, despite probable differences in the diet. Very limited data are available in the literature for camel. In the rare references, it was observed that there was a higher proportion of phosphorus [two times], magnesium [two times], and overall calcium [ten times] in camel rumen fluids than in cattle [80]. Among trace elements, only copper was determined in the above reference: 1.28 µg/mL on average, i.e., a lower mean value than in our result [3.66 µg/mL]. The mineral concentrations in rumen fluids were susceptible to change in the case of diseases such as trypanosomiasis, acidosis, indigestion, or frothy bloating [81]. Without links to region or season, it was possible to identify two mineral patterns [RF1 and RF2], with almost twice-higher concentrations of calcium, magnesium, phosphorus, and manganese in the second pattern. This indicates significant individual variability within a defined region, but such variability cannot be explained in the absence of data on the camel diet. The role of digestive diseases cannot be ignored.

### 4.5. Mineral Patterns in Whole Blood

Usually, minerals are determined in plasma or serum rather than whole blood, except sometimes iron and selenium [3,69]. As the red blood cells can contain a part, more or less important, of those minerals, the comparison with data focused on plasma or serum content is questionable. In our study, a regional effect occurred, with the samples from the Dammam region having more major elements [Ca, Mg, and P] and less trace elements [Co, Cu, I, and Mn] except Se and Zn, compared to the Riyadh region. This regional difference helps to explain two patterns: one with high trace elements and low main minerals [WB1], and the other with the opposite [WB3]. Such an oppositional trace element or major element in whole blood has no clear biological explanation.

### 4.6. The Particular Case of Heavy Metals

Most of the time, there was no high variability [seasonal or regional] of the heavy metals [Cd and Pb] in the different tissues. Heavy metals are not biological elements, but their presence, clearly linked to contamination, notably in meat and liver consumed by humans, could have a deleterious effect on human health. On average, the values of lead and cadmium in liver and meat in our survey were comparable or slightly above those reported by Sharkawy et al. [82] in Egypt, Asli et al. [39] in Iran, and Chafik et al. [72] in Morocco. In contrast, similar lead [2.01–3.21 µg/g] and cadmium values [0.83–0.91 µg/g] were observed in camel meat from Algeria [83]. In Oman, a lower lead content in camel meat was also reported at 0.15 μg/g [68]. A recent study in Nigeria [84] found that lead levels in liver ranged from 0.35 to 1.17 µg/g and in meat from 0.11 to 0.20 µg/g on average across two slaughterhouses. For cadmium, the same authors found levels from 0.07 to 0.40 µg/g [liver] and from 0.01 to 0.05 µg/g [meat] in the same abattoirs. In Saudi Arabia, recently, contents in the liver were found to be 0.94 ± 0.37 µg/g for lead and 0.078 ± 0.061 µg/g for cadmium [49]. In contrast, Al-Perkhdri [40] found higher levels of cadmium in the liver [3.61–4.58 g/g] and muscle [3.70–4.17 g/g] in the Kirkuk region [Iraq], despite the fact that those areas were contaminated. Liver concentrations were always higher than in muscle [74]. In any case, our mean value in liver [0.40 µg/g] was below the limit of the EU standard for cadmium [0.50 µg/g], while in meat and liver, lead contents [1.62 and 2.57 µg/g, respectively] were above the limit of the EU standard [0.10 and 0.20 µg/g, respectively]. It is important to note that such high values could impact the health of regular consumers of camel meat in Saudi Arabia.

### 4.7. The Relationships between Mineral Concentrations in Different Organs 

The mean mineral concentration in one organ is highly correlated to the mean mineral concentrations in the other organs, allowing us to consider the global mineral status of the animals: the increase or decrease of minerals in relation to feeding or health conditions is affecting more or less all the organs. In another point of view, all the mean values of minerals were correlated together except calcium. Calcium’s status, along with its specific hormonal regulation, could explain its relative "independence" from the only source of supply via feeding [85].

## 5. Conclusions

Collecting samples in slaughterhouses gives only a photographic record of the mineral status of different regions but cannot be supported by hypotheses on the quality of the diet. The regional differences observed in our study have a complex origin. Indeed, the organs represent a way for the storage of some minerals [notably trace elements] but are also the reflection of metabolism using minerals [notably major minerals and trace elements playing a role in metallo-enzymes]. Therefore, both major and minor minerals could be highly variable in the body according to the physiological status of the animals, their nutritional state, the quality of their diet, and the mineral status of the forages in relation to the soil components. Furthermore, the state of one’s health could play a role. Further investigations could be done by gathering more information on the breeding conditions of the camels.

## 6. Recommendation

Intensive research is needed to investigate the minerals’ biological mechanisms in camels under different environmental conditions and production systems using appropriate techniques such as mineral isotopes. Moreover, developing supplementation programs for camels according to seasons, region, and physiological status is highly recommended to avoid trace mineral deficiencies.

## Figures and Tables

**Figure 1 animals-12-03343-f001:**
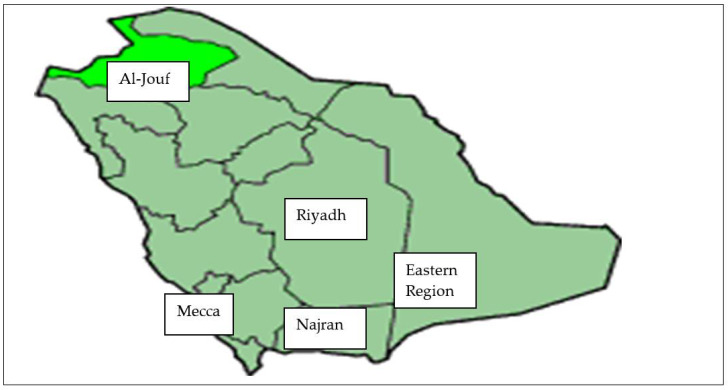
The map shows the five regions of this study. Central Region [Al Riyadh]; Eastern Region [Dammam]; Western Region [Mekka]; Southern Region [Najran]; and Northern Region [Al Jouf].

**Figure 2 animals-12-03343-f002:**
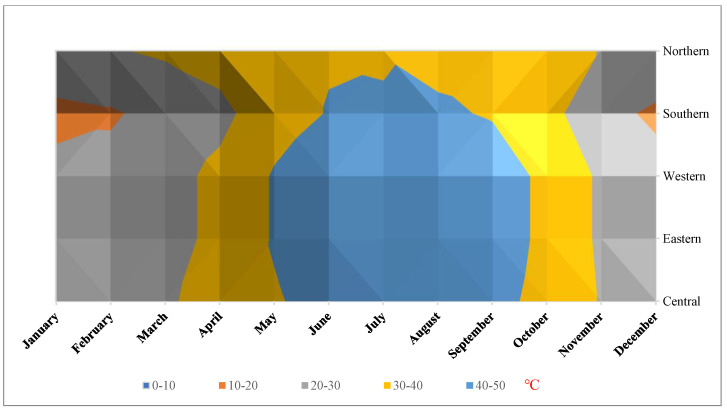
Graphs show the maximum and minimum temperatures (°C) in the different regions selected for this study: Central Region [Riyadh]; Western Region [Mekka]; Eastern Region [Dammam]; Southern Region [Najran]; and Northern Region [AlJouf]. General Authority for Statistics, GAS, [22].

**Figure 3 animals-12-03343-f003:**
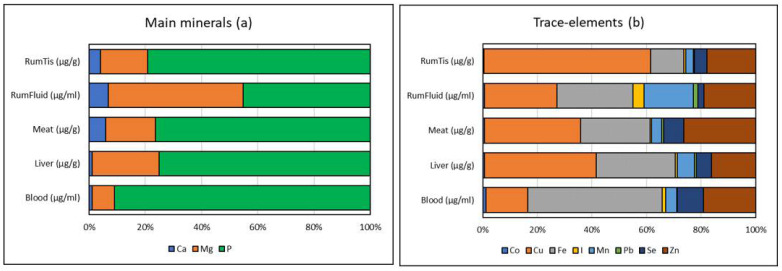
Proportion [in %] of different main minerals (**a**) and trace elements (**b**) in whole blood and organs of camel collected in five Saudi Arabian regions and at two seasons.

**Figure 4 animals-12-03343-f004:**
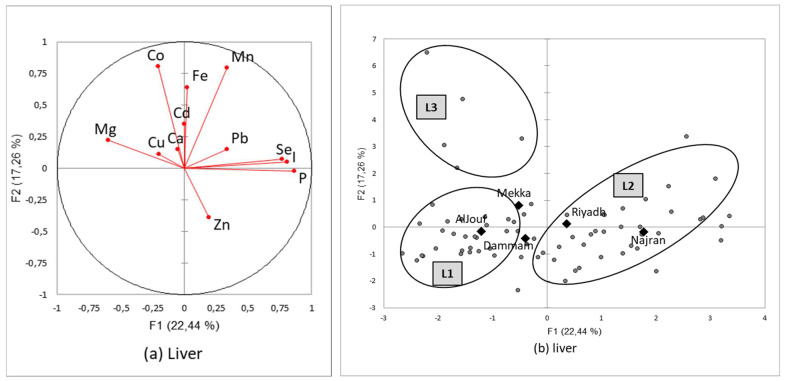
Correlation circles (**a**) for the two first factors of the Principal Components Analysis demonstrating the relationships between minerals in camel liver, and the main factorial plan (**b**) with representation of the three mineral patterns of the liver sampled in Saudi slaughterhouses from five regions and during two seasons (squares represents the mean value for each region and dots the different liver sample values).

**Figure 5 animals-12-03343-f005:**
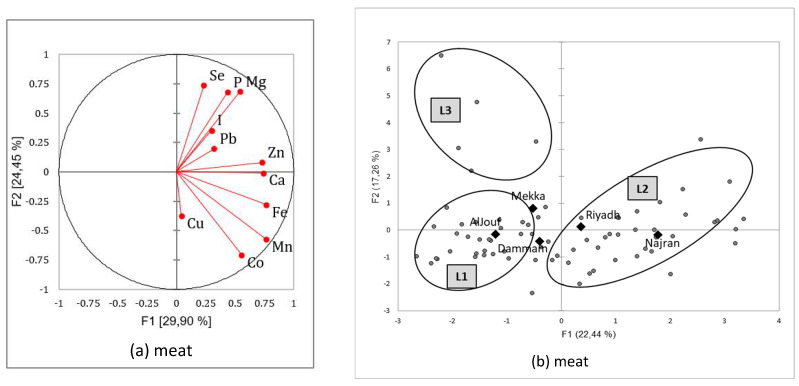
Correlation circles (**a**) for the two first factors of the Principal Components Analysis showing the relationships between minerals in camel meat, and main factorial plan (**b**) with representation of the three mineral patterns of the meat sampled in Saudi slaughterhouses from five regions and at two seasons (squares represents the mean value for each region and dots the different meat sample values).

**Figure 6 animals-12-03343-f006:**
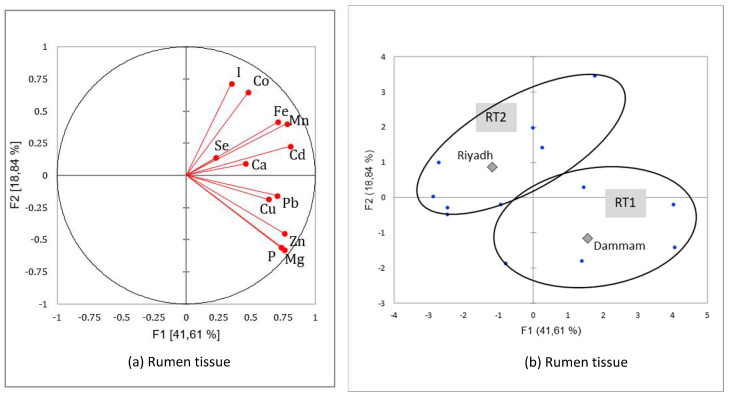
Correlation circles (**a**) for the two first factors of the Principal Components Analysis demonstrating the relationships between minerals in camel rumen tissue and main factorial plan (**b**) with representation of the two mineral patterns of the rumen tissue sampled in two regions of Saudi slaughterhouses at two seasons (squares represents the mean value for each region and dots the different rumen tissues sample values).

**Figure 7 animals-12-03343-f007:**
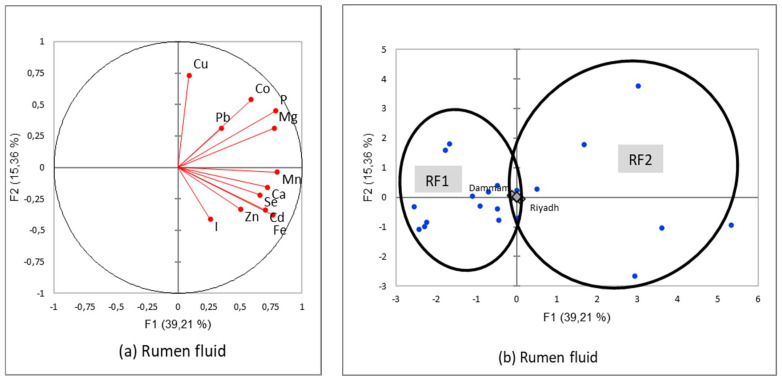
Correlation circles (**a**) for the two first factors of the Principal Components Analysis demonstrating the relationships between minerals in camel rumen fluid and the main factorial plan (**b**) with representation of the two mineral patterns of the rumen fluid sampled in two regions of Saudi slaughterhouses at two seasons (squares represents the mean value for each region and dots the different rumen fluid sample values).

**Figure 8 animals-12-03343-f008:**
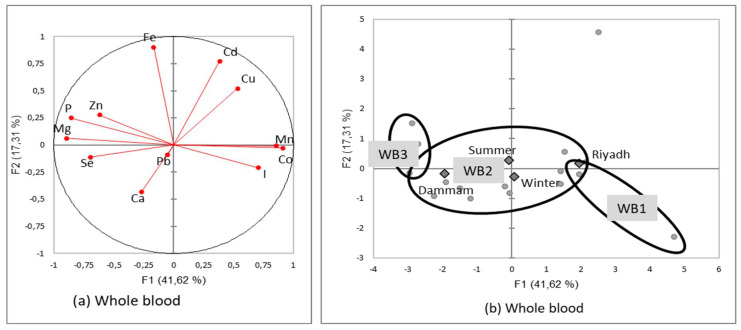
Correlation circles (**a**) for the two first factors of the Principal Components Analysis demonstrating the relationships between minerals in camel whole blood and the main factorial plan (**b**) with representation of the two mineral patterns of the whole blood sampled in two regions of Saudi slaughterhouses at two seasons (squares represents the mean value for each region and season and dots the different whole blood sample values).

**Table 1 animals-12-03343-t001:** The mineral concentration [ppb] in representative soil * samples from the five regions.

Region	Co	Cu	Cd	Pb	Se	Fe	I	Mn	Zn
Central	0.022	0.04	0.003	0.0002	0.232	2.18	1.36	0.027	0.021
Eastern	0.011	0.05	0.003	0.0005	0.109	1.01	0.87	0.055	0.016
Western	0.035	0.05	0.002	0.0004	0.276	0.29	1.34	0.027	0.012
Southern	0.004	0.014	0.002	0.0007	0.014	0.24	1.05	0.014	0.004
Northern	0.007	0.015	0.004	0.0006	0.090	0.49	0.85	0.0425	0.006

***** Soil samples were collected from each region, different sites, and seasons from the same area of camel grazing. The soil was excavated to a depth of 100 cm, and five representative samples of the sequence of the soil horizons were taken.

**Table 2 animals-12-03343-t002:** The mineral concentration [µg/mL] of drinking water * for camels from the five regions.

Region	Co	Cu	Cd	Pb	Se	Fe	I	Mn	Zn
Central	0.013	0.035	0.002	0.0001	0.233	2.175	1.351	0.029	0.019
Eastern	0.007	0.045	0.003	0.0002	0.277	0.320	1.605	0.019	0.005
Western	0.012	0.052	0.002	0.0004	0.109	1.012	0.805	0.055	0.016
Southern	0.004	0.050	0.0019	0.0005	0.070	0.240	0.056	0.034	0.003
Northern	0.008	0.019	0.0017	0.0003	0.0923	0.645	0.848	0.066	0.007

* Composite sample from a series of 100 mL water samples collected from different water sources that were consumed by camels in each region and season and mixed before a representative sample was drawn in a sterilized container.

**Table 3 animals-12-03343-t003:** Distribution of the samples according to seasons and regions.

Regions	Riyadh	Mekka	Dammam	Najran	Al-Jouf
Summer	20	41	23	7	16
Winter	30	21	9	6	3
Organs	Blood	Liver	Meat	Rum Fluid	Rum Tis
Summer	8	53	31	9	6
Winter	8	20	22	11	8

Rum Fluid = Rumen Fluid; Rum Tis = Rumen Tissues.

**Table 4 animals-12-03343-t004:** Mean values of minerals in the different camel organs and fluids.

Organ/Mineral	Ca	Cd	Co	Cu	Fe	I	Mg	Mn	P	Pb	Se	Zn
Blood [µg/mL]	1.08	0.03	0.29	3.78	12.07	0.310	8.33	0.97	94.77	0.06	2.37	4.67
Liver [µg/g]	69.89	0.41	2.41	150.50	105.61	2.71	1586.03	23.28	4983.73	2.57	20.56	59.10
Meat [µg/g]	216	0.215	1.15	67.05	48.40	1.07	651.53	6.74	2813.52	1.62	13.80	50.08
Rum Fluid [µg/mL]	19.60	0.025	0.10	3.66	3.82	0.57	136.04	2.47	128.57	0.23	0.31	2.60
Rum Tis [µg/g]	27.38	0.125	0.54	68.55	13.84	0.68	112.23	3.25	527.90	0.42	5.18	19.10

**Table 5 animals-12-03343-t005:** Mean values of minerals in camel liver according to region and season [µg/g wet weight].

Element	Ca	Cd	Co	Cu	Fe	I	Mg	Mn	P	Pb	Se	Zn
Season												
Summer	54.74	0.44	2.23	135.62	104.61	2.81	2171.69	24.18	4749.6	2.6	21.36	58.79
Winter	98.22	0.37	2.61	95.03	100.5	2.96	1628.84	24.63	5699.8	2.15	18.83	58.77
*p* value<	* **0.05** *	*NS*	*NS*	*NS*	*NS*	*NS*	*NS*	*NS*	* **0.019** *	*NS*	*NS*	*NS*
Region												
Riyadh	41.99 ^a^	0.28	3.17	137.83	89.45	2.57	559.35 ^a^	29.98 ^b^	5541.36 ^a^	3.38	16.47 ^a^	54.46
Mekka	65.69 ^ab^	0.50	1.81	181.30	104.5	2.57	1607.77 ^bc^	17.05 ^a^	4906.14 ^a^	1.58	16.38 ^a^	63.61
Dammam	153.43 ^b^	0.49	3.94	115.83	120.10	2.58	2185.54 ^c^	29.07 ^b^	4509.83 ^a^	3.48	19.45 ^a^	58.52
Najran	36.88 ^a^	0.37	1.95	68.28	86.47	2.79	4291.38 ^d^	23.36 ^ab^	4291.38 ^a^	1.45	5.31 ^a^	64.79
Al-Jouf	84.42 ^ab^	0.38	1.24	73.37	112.33	3.92	857.30 ^ab^	22.55 ^ab^	6874.76 ^b^	2.17	42.86 ^b^	52.50
*p* value<	* **0.004** *	*NS*	*NS*	*NS*	*NS*	*NS*	* **0.0001** *	* **0.007** *	* **0.003** *	*NS*	* **0.003** *	*NS*

Value within column followed with different superscripts are significantly differ. *p* ≤ 0.05 considered significantly differ between means. NS = Not Significant. All significant values are in bold

**Table 6 animals-12-03343-t006:** Mean values of minerals in camel meat according to region and season [µg/g wet weight].

Element	Ca	Cd	Co	Cu	Fe	I	Mg	Mn	P	Pb	Se	Zn
	Season											
Summer	195	0.159	0.57	48.61	47.06	1.15 ^b^	675.11	5.43	2922.16	1.76	13.57	50.42
Winter	235	0.273	1.61	75.75	51.24	0.92 ^a^	661.39	7.77	2910.31	1.28	14.83 ^a^	45.48
*p* value<	*NS*	*0.0219*	* **0.024** *	*NS*	*NS*	* **0.029** *	*NS*	*NS*	*NS*	*NS*	*NS*	*NS*
	Region											
Riyadh	236	0.118 ^b^	0.32 ^a^	20.01	45.85 ^a^	1.24 ^b^	788.63 ^b^	4.30 ^a^	3302.93 ^b^	1.30	16.76 ^bc^	46.14 ^ab^
Mekka	131	0.138 ^b^	0.49 ^ab^	87.69	32.36 ^a^	0.72 ^a^	630.05 ^ab^	2.94 ^a^	3039.82 ^ab^	1.12	10.42 ^ab^	34.94 ^a^
Dammam	212	0.130 ^b^	1.60 ^bc^	116.96	33.27 ^a^	1.11 ^b^	561.26 ^a^	6.01 ^a^	2371.93 ^a^	1.75	12.52 ^ab^	53.58 ^b^
Najran	240	0.412 ^a^	2.07 ^c^	51.05	82.47 ^b^	0.99 ^ab^	595.32 ^ab^	13.07 ^b^	2373.02 ^a^	1.43	4.60 ^a^	56.38 ^b^
Al-Jouf	256	0.213 ^b^	0.99 ^abc^	35.18	51.80 ^a^	1.11 ^b^	765.99 ^b^	6.68 ^a^	3493.48 ^b^	2.00	26.67 ^c^	48.73 ^b^
*p* value<	*NS*	* **0.0001** *	* **0.049** *	*NS*	* **0.002** *	* **0.048** *	* **0.048** *	* **0.008** *	* **0.004** *	*NS*	* **0.001** *	* **0.005** *

Value within column followed with different superscripts are significantly. *p* ≤ 0.05 considered significantly differ between means. NS = Not Significant; All significant values are in bold.

**Table 7 animals-12-03343-t007:** Mean values of minerals in camel rumen tissues according to region and season [µg/g wet weight].

Element	Ca	Cd	Co	Cu	Fe	I	Mg	Mn	P	Pb	Se	Zn
Season												
Summer	33.77	0.13	0.71	70.18	15.62	0.80	128.07	3.93	563.21	0.47	5.2	20.15
Winter	23.00	0.12	0.39	73.75	12.58	0.60	116.88	2.82	570.35	0.39	5.04	20.41
*p* value<	*NS*	*0.038*	*NS*	*NS*	*NS*	*NS*	*NS*	*NS*	*NS*	*NS*	*NS*	*NS*
Region												
Dammam	30.10	0.13	0.45	97.64	14.41	0.66	188.57	3.67	842.51	0.48	4.93	22.51
Riyadh	26.68	0.12	0.66	46.29	13.79	0.73	56.38	3.07	291.05	0.38	5.40	18.05
*p* value<	*NS*	*NS*	*NS*	*NS*	*NS*	*NS*	* **0.0001** *	*NS*	* **0.0001** *	*NS*	*NS*	* **0.001** *

*p* ≤ 0.05 considered significantly differ between means. NS = Not Significant. All significant values are in bold.

**Table 8 animals-12-03343-t008:** Mean values of minerals in camel rumen fluid according to region and season [in µg/L].

Element	Ca	Cd	Co	Cu	Fe	I	Mg	Mn	P	Pb	Se	Zn
Season												
Summer	20.68	0.02	0.10	4.11	4.16	0.63	129.47	2.92	115.39	0.25	0.30	2.39
Winter	18.76	0.03	0.10	3.22	3.56	0.55	140.97	2.07	139.13	0.21	0.33	2.74
*p* value<	*NS*	*NS*	*NS*	*NS*	*NS*	*NS*	*NS*	*NS*	*NS*	*NS*	*NS*	*NS*
Region												
Dammam	19.49	0.02	0.09	4.06	3.75	0.45	137.70	2.72	128.56	0.24	0.32	2.77
Riyadh	19.96	0.02	0.11	3.27	3.96	0.73	132.75	2.27	125.95	0.22	0.31	2.36
*p* value<	*NS*	*NS*	*NS*	*NS*	*NS*	*NS*	*NS*	*NS*	*NS*	*NS*	*NS*	*NS*

*p* ≤ 0.05 considered significantly differ between means. NS = Not Significant.

**Table 9 animals-12-03343-t009:** Mean values of minerals in camel whole blood according to region and season [in µg/L].

Element	Ca	Cd	Co	Cu	Fe	I	Mg	Mn	P	Pb	Se	Zn
Season												
Summer	1.11	0.03	0.30	3.20	14.27	0.47	8.75	1.03	94.08	0.06	2.50	4.86
Winter	1.07	0.03	0.27	4.35	9.87	0.15	7.90	0.91	95.45	0.06	2.24	4.40
*p* value<	*NS*	*NS*	*NS*	*NS*	*NS*	*NS*	*NS*	*NS*	*NS*	*NS*	*NS*	*NS*
Region												
Dammam	1.21	0.03	0.04	1.95	12.54	0.11	10.20	0.35	111.39	0.06	2.84	4.98
Riyadh	0.96	0.03	0.54	5.60	11.61	0.51	6.45	1.58	78.14	0.06	1.89	4.35
*p* value<	* **0.025** *	*NS*	* **0.000** *	* **0.012** *	*NS*	* **0.041** *	* **0.0001** *	* **0.001** *	* **0.009** *	*NS*	* **0.038** *	* **0.038** *

*p* ≤ 0.05 considered significantly differ between means. NS = Not Significant. All significant values are in bold.

**Table 10 animals-12-03343-t010:** Mean values of minerals in camel liver according to the patterns based on Ascending Hierarchical Classification patterns [µg/g wet weight].

	Ca	Cd	Co	Cu	Fe	I	Mg	Mn	P	Pb	Se	Zn
Pattern L1	76.46	0.36	1.97 ^b^	125.03	82.21 ^b^	1.87 ^b^	2080.92 ^ab^	17.80 ^b^	3551.40 ^a^	1.56	9.66 ^b^	50.42 ^b^
Pattern L2	65.04	0.47	1.51 ^b^	157.44	100.38 ^b^	3.40 ^a^	1053.35 ^b^	24.34 ^b^	6109.97 ^b^	3.45	29.80 ^a^	66.36 ^a^
Pattern L3	72.00	0.68	12.15 ^a^	237.59	278.65 ^a^	1.92 ^b^	3076.18 ^a^	45.51 ^a^	3994.96 ^a^	1.23	7.72 ^b^	49.72 ^b^
*p* value<	*NS*	*NS*	* **0.0001** *	*NS*	* **0.000** *	* **0.000** *	* **0.000** *	* **0.0001** *	* **0.0001** *	*NS*	* **0.0001** *	* **0.001** *

Value within column followed with different superscripts are significantly differ. NS = Not Significant. All significant values are in bold.

**Table 11 animals-12-03343-t011:** Mean values of minerals in camel meat according to the patterns issued from Ascending Hierarchical Classification [µg/g wet weight].

	Ca	Co	Cu	Fe	I	Mg	Mn	P	Pb	Se	Zn
Pattern M1	150 ^a^	0.57 ^a^	46.39	34.39 ^a^	0.94 ^a^	508.0 ^a^	3.61 ^a^	2286.68 ^a^	1.02	8.06 ^a^	40.64 ^a^
Pattern M2	241 ^b^	0.52 ^a^	47.23	48.03 ^a^	1.28 ^b^	814.68 ^b^	5.05 ^a^	3472.45 ^b^	2.18	24.64 ^b^	53.68 ^b^
Pattern M3	308 ^b^	3.77 ^b^	154.12	80.02 ^b^	0.94 ^a^	624.69 ^a^	17.17 ^b^	2588.81 ^a^	1.74	3.68 ^a^	63.27 ^c^
*p* value<	* **0.000** *	* **0.0001** *	*NS*	* **0.001** *	* **0.004** *	* **0.0001** *	* **<0.0001** *	* **0.0001** *	*NS*	* **0.0001** *	* **0.0001** *

Value within column followed with different superscripts are significantly differ. NS = Not Significant. All significant values are in bold.

**Table 12 animals-12-03343-t012:** Mean values of minerals in camel rumen tissue according to the patterns issued from Ascending Hierarchical Classification [µg/g wet weight].

	Ca	Cd	Co	Cu	Fe	I	Mg	Mn	P	Pb	Se	Zn
Pattern RT1	30.10	0.13	0.45	97.64	14.41	0.66	188.57	3.67	842.51	0.48	4.93	22.51
Pattern RT2	25.34	0.12	0.62	46.73	13.41	0.71	54.98	2.93	291.94	0.37	5.37	18.08
*p* value>	*NS*	*NS*	*NS*	*NS*	*NS*	*NS*	* **0.0001** *	*NS*	* **0.0001** *	*NS*	*NS*	* **0.000** *

*p* ≤ 0.05 considered significantly differ between means. NS = Not Significant. All significant values are in bold.

**Table 13 animals-12-03343-t013:** Mean values of minerals in camel rumen fluid according to the patterns issued by Ascending Hierarchical Classification [µg/L].

	Ca	Cd	Co	Cu	Fe	I	Mg	Mn	P	Pb	Se	Zn
Pattern RF1	16.29	0.02	0.08	3.97	3.42	0.44	80.62	0.91	85.19	0.21	0.29	2.51
Pattern RF2	22.92	0.03	0.11	3.35	4.22	0.70	191.47	4.04	171.93	0.25	0.34	2.69
*p* value<	* **0.026** *	*NS*	*NS*	*NS*	*NS*	*NS*	* **0.0001** *	* **0.004** *	* **0.0001** *	*NS*	*NS*	*NS*

*p* ≤ 0.05 considered significantly differ between means. NS = Not Significant. All significant values are in bold.

**Table 14 animals-12-03343-t014:** Mean values of minerals in camel whole blood based on Ascending Hierarchical Classification patterns [µg/L].

	Ca	Cd	Co	Cu	Fe	I	Mg	Mn	P	Pb	Se	Zn
Pattern WB1	1.21	0.02	0.83 ^b^	2.76	8.73	1.16 ^b^	5.91 ^a^	2.28 ^b^	50.95 ^a^	0.08	1.42	4.72 ^ab^
Pattern WB2	1.01	0.02	0.26 ^a^	4.36	11.42	0.21 ^a^	7.89 ^a^	0.89 ^a^	90.73 ^b^	0.06	2.40	4.45 ^a^
Pattern WB3	1.26	0.02	0.03 ^a^	2.31	16.70	0.12 ^a^	11.55 ^b^	0.36 ^a^	138.77 ^c^	0.06	2.89	5.43 ^b^
*p* value<	*NS*	*NS*	* **0.011** *	*NS*	*NS*	* **0.002** *	* **0.002** *	* **0.019** *	* **0.0001** *	*NS*	*NS*	* **0.045** *

Value within column followed with different superscripts are significantly NS = Not Significant. All significant values are in bold.

**Table 15 animals-12-03343-t015:** Matrix of correlation of the mean mineral values between the different biological samples.

Variables	Mean Liver	Mean RumenF	Mean RT	Mean Meat	Mean Blood
Mean Liver	**1**	**0.847**	**0.988**	**0.996**	**0.964**
Mean RumenF	**0.847**	**1**	**0.777**	**0.799**	**0.688**
Mean RT	**0.988**	**0.777**	**1**	**0.993**	**0.980**
Mean Meat	**0.996**	**0.799**	**0.993**	**1**	**0.981**
Mean blood	**0.964**	**0.688**	**0.980**	**0.981**	**1**

Bold values are significant.

## Data Availability

The data presented in this study are available on request from the Corresponding author.

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
