# Peer review of "Regional and Seasonal Variability of Mineral Patterns in Some Organs of Slaughtered One-Humped Camels [Camelus dromedarius] from Saudi Arabia"

_animals, 2022, doi:10.3390/ani12233343_

Round 1

Reviewer 1 Report

As a general comment, I find that this is an interesting manuscript in regard to increasing meat demand for human consumption. However, the interesting results, need to be discussed according to the objective of safe human food. Actually, the discussion is only based to explain the source of difference and the bilological significance of the results. While this is fine, the authors need also to adress for example the high Pb an Cd, in some instances, in relation to human food safety. Those heavy metals are involved in human health problems. After Reading this manuscript, I am having concerns about eating liver or Camel meat in SA. These concerns were not adressed on a human perspective. Thus, the discussion needs to be reviewed accordingly.

Also, the discussion should not be treated in "silo" type of discussion as the authors did. While for the result section, it is fine to subdivise in sub-sections for blood, rumen fluid, liver, meat etc., but the discussion should make a link between all these and discuss the results as a whole linked process. The link needs to be done bewtween what is in the soil, water, enventually in rumen fluid, in blood, , liver, in tissus and  meat. Every thing is linked (soil/water/plant/meat). This particular manuscript is a good example of that link, but it is not discussed in this way.

SPECIFIC COMMENTS:

L64: remove "that"

uniformize table 1 and 2 to keep the same order in elements

Heavy metal do not appear in the abstract. Why? add the information

L131-139. Method for blood samples analysis not clearly provided

table 3; 6 and 11: Cd is not provided in the results. PLease provide the results

table 7: edit "0,13b"

table 9: no "bold values" are shown contrary to what is written in footnote

In the MM: how were water and soil collected and analysed?

L479: what is considered "normal concentration" and cite a reference

L486-488: uniformize units

L572: interesting hypothesis for parasitism. If this is a problem in SA, why fecal egg count was not performed or other parasite determination?

L575-579: the authors wrote that Se deficiency is widely observed, but that there is no hepatic Se reference values. What about other reference values? Blood Se, blood or plasma GPx? Where the camels studied Se deficent in the present study? 

Author Response

Dear reviewer,

Thank you for your valuable comments.

At the begging I like to explain the main purpose of this study. A great shortage in camels nutrition data in the semi-arid area especially Saudi Arabia. So it is very hard to establish any supplementation programs to improve the their productivity. So, this study was conducted to establish very important information regarding the minerals status of camels in different regions in Saudi Arabia. This will be a very solid background to end up with a conclusion and recommendation. Furthermore, human health is also a concern when related to human health since meat and liver are highly consumed by human and may affect health

  • All comments suggested were considered especially in the discussion pat.
  • All specific comments in different lines were corrected.
  • Regarding the fecal eggs, we didn’t considered since we only focused on mineral status and linked to poor camel performance.
  • Regarding Se deficiency, this study was just a survey to obtain a data regarding the status and possibility of supplementation in case of deficiencies. Se deficiency reported in previous studies under different conditions.

Thanks again,

Best regards,

Prof. Mutassim M. Abdelrahman

Reviewer 2 Report

Dear Authors,

Thank you for submitting this manuscript that explores the mineral patterns in camel tissues in Saudi Arabia. This is an interesting study and it could have some wider impact in terms of human health.

At current however, there seem to be some large revisions required in the manuscript to ensure the work is scientifically robust. I have attached the PDF version of the manuscript with specific comments. Additionally, please consider the following points: 

1. Background. The explanation to the study is unclear. Is this for human health or camel health? Why is the difference between winter and summer so important? At current this is not clearly explained in the manuscript.

2. Slaughter. Would the animals have been slaughtered anyway? What was done with the carcasses following sampling?

3. Background of animals. The animal's background and ages are unclear. lack of information here brings in a lot of variability and also limits the repeatability of your study. Please state clearly where the animals came from, what they were fed and how they were kept. provide details including their ages and strains, if known. These points all impact repeatability.

4. Wording. The manuscript is grammatically confusing and as such, a full proof read is needed, ideally by a professional copyeditor.

5. Test choice. Please see points relating to the choice of test (normal distribution) and apply a correction factor to reduce your risk of type 1 error. Please provide exact p values and test statistics consistently throughout the work.

6. Reference list. There are numerous errors in formatting. Please see the Animals author guidelines and revise thoroughly.

If all points are covered, the manuscript should be in a position for consideration.

Author Response

Dear reviewer,

Thank you for your valuable comments.

At the begging I like to explain the main purpose of this study. A great shortage in camels nutrition data in the semi-arid area especially Saudi Arabia. So it is very hard to establish any supplementation programs to improve the their productivity. So, this study was conducted to establish very important information regarding the minerals status of camels in different regions in Saudi Arabia. This will be a very solid background to end up with a conclusion and recommendation. Furthermore, human health is also a concern when related to human health since meat and liver are highly consumed by human and may affect health.

  • We considered all specific comments that provided in the attached PDF file.
  • 1 and 2. This study targeted the mineral status of the slaughtered camels raised in a semi-arid areas in Saudi Arabia as a base line for their status, so, their effect on human health can be achieved since the meat and liver are mainly consumed from the slaughterhouse.
  • Season difference were reported in many previous study and effect on mineral intake, digestion and utilization were affected by seasons.
  • The animal background is not easy to have all the details because we collected the samples from the slaughterhouse, but we try to obtain a homogeneous camels especially in term of age (!.5 to 5 years old males).
  • We considered the wording as requested.
  • We revised your comments regarding the statistics.
  • All reference s were revised and corrected according to animals’ journal guidelines.

Thanks again,

Best regards,

Prof. Mutassim M. Abdelrahman

Round 2

Reviewer 1 Report

in table 6, the column of Cd show not data. Add the data.

Author Response

Dear reviewer,

I appreciate your comment and I added the data of Cd in table 6 and in the text.

I appreciate your valuable comments and revision.

Best regards, Prof. Mutassin M. Abdelrahman

Reviewer 2 Report

Dear Authors,

Many thanks for submitting this revised version of the manuscript for review. You have taken into account the feedback provided on the initial review of the paper. You have also shown clearly where changes have been made to the work. The developments to the manuscript have resulted in a more robust paper overall. In light of the revisions, the paper is now in a much better position for consideration.

Author Response

Dear reviewer,

Thanks and best regards,

Prof. Mutassim M. Abdelrahman